# Cooperative tumour cell membrane targeted phototherapy

Heegon Kim[1,2], Junsung Lee[2,3], Chanhee Oh[1,2] & Ji-Ho Park[1,2,3,4,5]

The targeted delivery of therapeutics using antibodies or nanomaterials has improved the precision and safety of cancer therapy. However, the paucity and heterogeneity of identified molecular targets within tumours have resulted in poor and uneven distribution of targeted agents, thus compromising treatment outcomes. Here, we construct a cooperative targeting system in which synthetic and biological nanocomponents participate together in the tumour cell membrane-selective localization of synthetic receptor-lipid conjugates (SR-lipids) to amplify the subsequent targeting of therapeutics. The SR-lipids are first delivered selectively to tumour cell membranes in the perivascular region using fusogenic liposomes. By hitchhiking with extracellular vesicles secreted by the cells, the SR-lipids are transferred to neighbouring cells and further spread throughout the tumour tissues where the molecular targets are limited. We show that this tumour cell membrane-targeted delivery of SR-lipids leads to uniform distribution and enhanced phototherapeutic efficacy of the targeted photosensitizer.

[1] Department of Bio and Brain Engineering, Korea Advanced Institute of Science and Technology (KAIST), Daejeon 34141, Korea. [2] KAIST Institute for Health Science and Technology, Korea Advanced Institute of Science and Technology (KAIST), Daejeon 34141, Korea. [3] Graduate School of Medical Science and Engineering, Korea Advanced Institute of Science and Technology (KAIST), Daejeon 34141, Korea. [4] Program of Brain and Cognitive Engineering, Korea Advanced Institute of Science and Technology (KAIST), Daejeon 34141, Korea. [5] KAIST Institute for the Nanocentury, Korea Advanced Institute of Science and Technology (KAIST), Daejeon 34141, Korea. Correspondence and requests for materials should be addressed to J.-H.P. (email: jihopark@kaist.ac.kr).

Targeted therapies acting on specific molecular targets in tumour microenvironments, such as monoclonal antibodies (including those conjugated to drugs) and small molecules, have been developed to overcome limitations of transitional chemotherapies[1]. Nanoscale materials decorated with targeting ligands have also been harnessed to encapsulate anti-cancer drugs and improve their tumour-targeting efficacy through the enhanced permeation and retention effect, and multivalent binding to tumour-associated targets[2,3]. However, these targeted therapies have often failed because the tumour distribution of molecular targets is intrinsically heterogeneous (different types of cell in the tumour microenvironment and different numbers of receptors expressed on these tumour cell variants)[4–6]. Recently, cooperative targeting system strategies have been proposed to amplify the tumour homing of therapeutic and imaging agents, regardless of the intrinsic receptors[7–11]. In such cooperative targeting systems, pre-administered functional agents generate either biological or artificial binding sites in tumours, and the altered tumour microenvironment is subsequently occupied by targeted agents. However, in most cases, the distribution of targeted agents followed the distribution of pre-administered agents within solid tumours. For example, the targeting of therapeutic agents following nanoparticle-based delivery of synthetic receptors (SRs) is restricted to cells in perivascular regions because the transport of nanoparticles carrying SRs is significantly hindered by physiological barriers in the tumour microenvironment, such as high interstitial fluid pressure and a dense collagen fibre matrix[12]. The resulting poor distribution of therapeutic agents in the tumour reduces the efficacy of anti-cancer treatments.

Extracellular vesicles (EVs) are known to mediate intercellular communication by transferring lipids, cytosolic proteins and RNA through membrane fusion[13–15]. They also play a supportive role in promoting tumour progression in that tumour-derived EVs deliver oncogenic signals to normal host cells[16,17]. Here, we seek to leverage their ability to transfer membrane-derived lipids between cells to distribute SR-lipid conjugates (SR-lipids) throughout tumour tissues and improve the therapeutic responses of membrane-targeted agents. Specifically, we design a cooperative tumour cell membrane targeting nanosystem to improve cancer therapy (Fig. 1). Synthetic liposomes engineered to fuse with plasma membranes (referred to here as fusogenic liposomes, FLs)[18,19] are used to deliver the SR-lipids efficiently to the plasma membranes of cells accessible from the vessels and then produce EVs packaging the SR-lipids for their transport from the cells. The SR-lipids then spread over multiple cell layers autonomously via EV-mediated intercellular transport and the therapeutic agents target the SRs on the cell surface throughout the entire tumour. In contrast, the SR-lipids that accumulate in the mononuclear phagocytic system, which is known to clear out most circulating nanomedicines[20,21], do not contribute to the binding of therapeutic agents due to their rapid intracellular uptake. We employ a biotin-streptavidin model system to verify our cooperative targeting nanosystem strategy because biotin-phospholipids as SR-lipids are transported via both FLs and EVs for the decoration of tumour cell membranes, and streptavidin (SA) carrying therapeutic molecules is small enough to diffuse into tumour tissues and find biotin-decorated membranes.

## Results

### Cell membrane-selective delivery of SR-lipids.
EVs are formed by naturally packaging cytosolic contents with the membrane of parental cells. Based on this mechanism of EV biogenesis, functional lipids in the membrane of parental cells could be further incorporated in the membrane of EVs secreted from the cells. Thus, we first investigated whether SR-lipids could be transferred efficiently to the membrane of tumour cells by fusion of the liposomal and plasma membrane. Highly cationic liposomes (CLs), which enter the cell rapidly via endocytosis by interacting with the plasma membrane, and PEGylated liposomes (PLs), which interact poorly with the plasma membrane, were also prepared alongside FLs for comparison. Biotin/fluorophore-lipids were incorporated into liposomal membranes at a molar ratio of 5% (biotin/fluorophore-liposome, Table 1). HeLa cells were treated with biotin-liposomes for 1 h and washed thoroughly. The cells were then incubated with fluorophore-conjugated SA (fluorophore-SA) to detect biotin on the cell surface. Confocal microscopy revealed that FLs delivered biotin-lipids throughout the plasma membrane most effectively (Fig. 2a). In contrast, the cells treated with CLs showed poor distribution of biotins on the surface, and PLs hardly delivered any biotin-lipids to the membrane. In fluorescence quantification, FLs and CLs delivered substantial amounts of biotin-lipids to the cells, while the amount was negligible for PLs. The level of such biotin moieties created on the cell surface decreased over time, with a half life of ~16 h (Fig. 2b). Confocal microscopy also showed that fluorophore-lipids localized on plasma membranes were translocated into intracellular membranes over time, reflecting the dynamic characteristics of membrane lipids[22] (Supplementary Fig. 1). Furthermore, the amount of cell surface biotins was controllable using the dose of biotin-FLs, without affecting cell viability (Supplementary Fig. 2).

### EV-mediated intercellular transfer of SR-lipids.
Next, we studied whether the SRs could be created on neighbouring cell surfaces via EV-mediated intercellular lipid transfer. Transwell experiments were performed to observe the intercellular transfer of SR-lipids (Supplementary Fig. 3). HeLa cells in the upper filter with 400-nm pores, treated with biotin-liposomes for 1 h, were co-incubated with fresh cells in the lower chamber for 4 h. The cells in the lower chamber were then incubated with fluorophore-SA to detect biotins on the cell surface. By confocal microscopy and fluorescence quantification, the highest level of biotins was detected on the lower chamber cell surface when FLs delivered biotin-lipids to the upper filter cells (Fig. 2c). Importantly, the amount of biotin-lipids on the lower chamber cell surface remained high until 12 h after membrane-selective delivery to the upper filter cells and then decreased over time (Fig. 2d). To prove that EVs primarily mediate the intercellular transfer of SR-lipids, EVs were removed from the supernatant collected from biotin-FL-treated cells using an established ultracentrifugation protocol for EV isolation[23], and then fresh cells were treated with the EV-depleted supernatant. The amounts of cell surface biotins were remarkably reduced when EVs were depleted from the medium (Fig. 2e). In addition, the biotin-lipid transfer to the lower chamber cells was significantly reduced by selective blocking the expression of Rab GTPases, which regulate exosome secretion[24], or ADP-ribosylation factor 6 (ARF6), which regulates microvesicle secretion[25], in the transwell filter cells (Fig. 2f). EV-mediated delivery of SR-lipids to neighbouring cells was also confirmed with the result showing that cellular uptake of EVs packaging fluorophore-lipids was significantly reduced in the presence of heparin, a known inhibitor of cellular uptake of EVs (ref. 26; Supplementary Fig. 4; Supplementary Methods).

To verify EV packaging of SR-lipids more directly, we first employed dimethyl amiloride (DMA), an inhibitor of exosome secretion[27]. DMA pretreatment reduced significantly both quantity and fluorescence signal of EVs produced from fluorophore-FL-treated cells (Fig. 2g and Supplementary Fig. 5; Supplementary Methods), indicating that exosomes packaged the

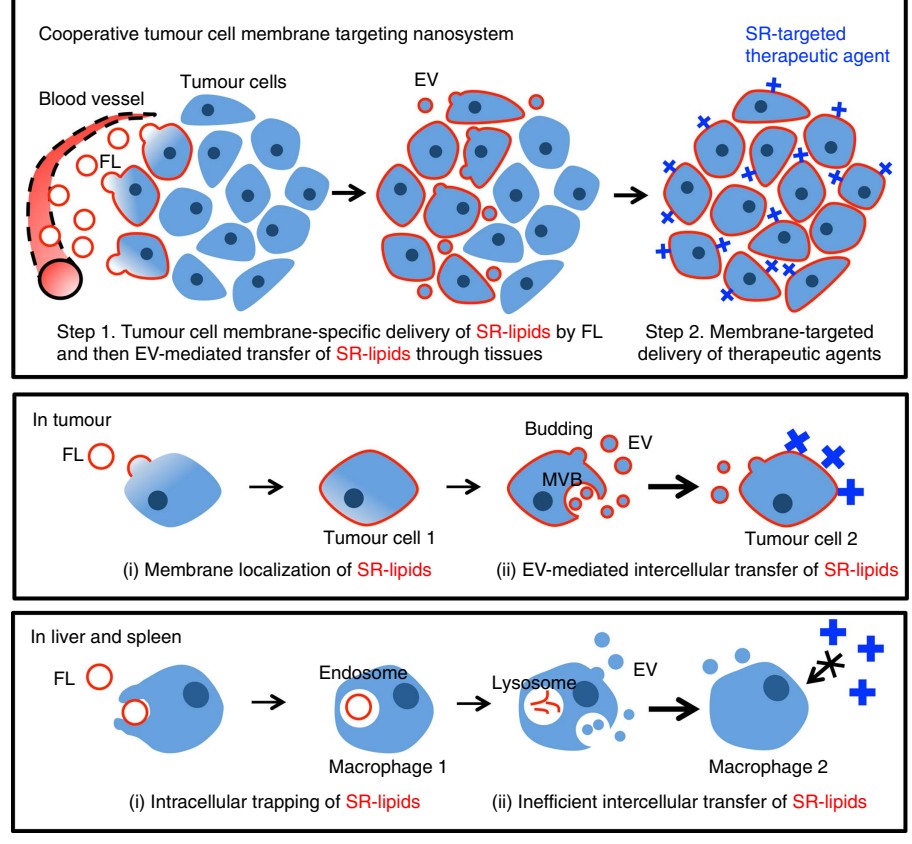

**Figure 1 | Schematic representation of cooperative tumour cell membrane targeting nanosystem.** In step 1, FLs are intravenously injected to deliver SR-lipid conjugates (SR-lipids, red) specifically to the plasma membranes of tumour cells in the perivascular regions. The SR-lipids are then incorporated into the membrane of EVs secreted from the cells, and transferred to the membranes of neighbouring cells via EVs. This active EV-mediated transfer of SR-lipids occurs through tumour tissues over 24 h after SR-lipid injection. In step 2, SR-targeted proteins (blue) that are capable of tumour penetration due to their relatively small size are intravenously injected to deliver therapeutic agents to the tumour cells pretreated with SR-lipids. Importantly, FLs deliver SR-lipids efficiently to the plasma membranes in tumours for subsequent EV-mediated spreading, while FLs taken up by macrophages in the mononuclear phagocyte system undergo lysosomal degradation, minimally exposing the SRs on the macrophage surface.

**Table 1 | Biological and physicochemical properties of materials used in this study.**

| Material | Lipid composition (molar ratio) | | | | Hydrodynamic size* (nm) | Surface charge† (mV) | Blood half-life‡ (h) |
|---|---|---|---|---|---|---|---|
| | **DMPC** | **PEG-PE** | **DOTAP** | **Biotin-PE/Alexa488-PE** | | | |
| FL | 71.2 | 3.8 | 20 | 5 | 126.5 | 15.4 | 2.2 |
| CL | 75 | 0 | 20 | 5 | 135.0 | 45.9 | – |
| PL | 91.2 | 3.8 | 0 | 5 | 121.1 | -10.3 | 16.1 |
| SA | – | – | – | – | 6.5 | -4.2 | 14.3 |

*Mean hydrodynamic sizes of the materials based on dynamic light scattering measurements ($n = 3$).
†Mean surface charges of the materials based on zeta-potential measurements ($n = 3$).
‡Blood half-lives of the materials obtained by fitting blood fluorescence to a single-exponential equation for a one-compartment open pharmacokinetic model ($n = 3$).

fluorophore-lipids delivered to the cell. We also quantified the percentage of EVs packaging fluorophore-lipids out of total EVs produced from the cells treated with liposomes containing fluorophore-lipids.

FL-treated cells produced EVs packaging fluorophore-lipids ($> 40\%$) more efficiently compared with CL- and PL-treated cells (4.4% and 1.7%, respectively; Fig. 2h). The presence of SR-lipids in the EVs was further confirmed with the results of sucrose gradient ultracentrifugation and transmission electron microscopy (Supplementary Figs 6 and 7; Supplementary Methods). The incorporation of SR-lipids was unlikely to alter

morphology, protein profile (including exosomal markers CD63 and CD9) and size of EVs (Supplementary Fig. 8; Supplementary Methods). These results suggest that selective delivery of SR-lipids into the plasma membrane is necessary for their efficient incorporation in the EV membrane because microvesicles and exosomes are produced by outward budding of plasma membrane and inward budding of endosomal membrane, respectively (Supplementary Fig. 9). These observations also support the notion that the EVs play an important role in mediating the intercellular transfer of SR-lipids. Furthermore, it is expected that the SR-lipids could penetrate multiple cell layers by

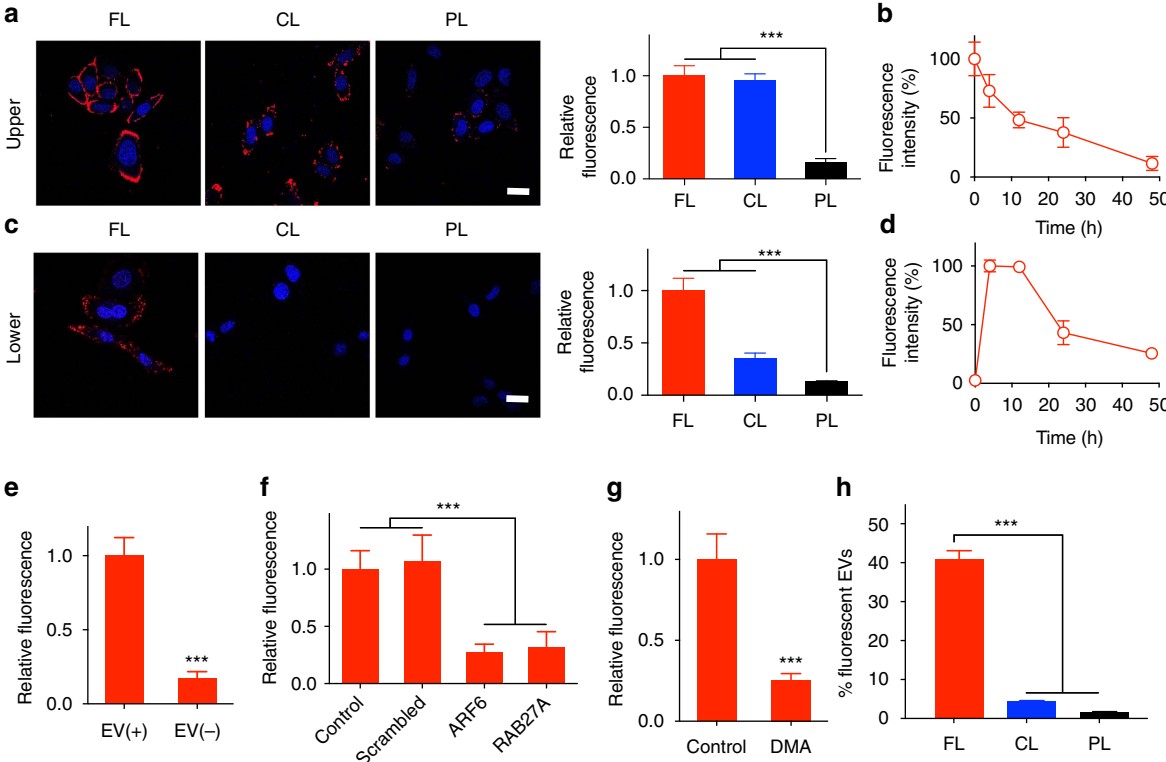

**Figure 2 | Cell membrane-selective delivery and EV-mediated intercellular transfer of SR-lipids.** (**a**) Confocal fluorescent microscopic images and fluorescence quantification of HeLa cells treated with fluorophore-streptavidin (SA; red) after liposome-mediated biotin-lipid delivery for 1 h. (**b**) Time-dependent fluorescence change of cells after FL-mediated biotin-lipid delivery in **a**. (**c**) Confocal fluorescent microscopic images and fluorescence quantification of HeLa cells in the lower chamber of a transwell. Cells in the upper filter with 400-nm pores, treated with biotin-liposomes for 1 h, were co-incubated with fresh cells in the lower chamber for 4 h. The cells in the lower chamber were then incubated with fluorophore-SA (red) and imaged. (**d**) Time-dependent fluorescence change of lower-chamber cells after FL-mediated biotin-lipid delivery to the transwell filter cells in **c**. (**e**) Fluorescence quantification of HeLa cells treated with fluorophore-SA after treatment with EV-containing ( + ) or EV-depleted ( − ) supernatant collected from the cells treated with biotin-FLs. (**f**) Fluorescence quantification of HeLa cells in the lower chamber treated with fluorophore-SAs 4 h after FL-mediated biotin-lipid delivery to siRNA (scrambled, ARF6 or RAB 27A)-pretreated cells in the upper transwell filter. (**g**) Fluorescence quantification of EVs produced from the cells treated with FLs containing fluorophore-lipids after DMA treatment. (**h**) Nanoparticle tracking analysis of EVs produced from the cells treated with liposomes containing fluorophore-lipids. Data are means ± s.e.m. (n ≥ 10, ***P < 0.001, one-way analysis of variance (ANOVA) with Tukey post test for **a** and **c**; n = 3 for **b** and **d**; n = 5; ***P < 0.001, Student's t-test for **e**,**f** and **g**; n = 3, ***P < 0.001, one-way ANOVA with Tukey's *post hoc* test for **h**). Scale bar represents 20 μm.

successive rounds of membrane fusion and subsequent EV secretion. Taken together, these results demonstrate that SR-lipids can be delivered selectively to the plasma membrane by a synthetic nanocomponent, FL and then autonomously transferred to the plasma membrane of neighbouring cells by a biological nanocomponent, EV.

**Tumour cell membrane-specific delivery of SR-lipids**. We next examined whether SR-lipids could be localized more selectively onto the plasma membrane of tumour cells than that of macrophages in the mononuclear phagocytic system. Fluorophore-lipids were used to observe the subcellular localization of SR-lipids delivered to cells. Tumour cells or macrophages were treated with fluorophore-FLs for 1 h and their plasma membranes and lysosomes were stained. Confocal microscopy revealed that fluorophore-lipids were highly colocalized with plasma membranes in tumour cells, whereas they were with lysosomes in macrophages (Fig. 3a and Supplementary Fig. 10). Next, we tested the time-dependent binding of targeted agents after the membrane-selective delivery of SR-lipids. Cells were treated with fluorophore-SA immediately or 4 h after FL-mediated biotin-lipid delivery. Membrane localization of fluorophore-SA was observed in tumour cells, even after the 4-h period following biotin-lipid delivery, while it was not observed at all in either of the macrophages (Fig. 3b). In

addition, the membrane-specific delivery of fluorophore-SA was inefficient in the endothelial cells and fibroblasts due to poor membrane fusion of biotin-FLs (Supplementary Fig. 11). These experiments demonstrate that the SR-lipids can be localized efficiently on the plasma membrane of tumour cells via FL-mediated membrane fusion, thus providing synthetic binding sites for targeted counterparts. In contrast, phagocytic cells such as macrophages are unlikely to expose the SRs on their surface because they take up the liposomes rapidly while minimizing their membrane fusion. Furthermore, the tumour cell-derived EVs interacted more efficiently with tumour cells compared with macrophages, fibroblasts and endothelial cells, indicating preferential localization of SR-lipids in the tumour cells in the tumour microenvironment (Supplementary Fig. 12; Supplementary Methods). Collectively, these results suggest that the SR decoration are more dominant on tumour cell surfaces than on other types of cells including macrophages, endothelial cells and fibroblasts, enabling the subsequent delivery of targeted agents in a tumour cell membrane-selective manner.

**Cooperative membrane-selective photocytotoxicty.** Since substantial membrane-specific spreading of SRs was observed, we proceeded to assess the therapeutic potential of membrane-targeted delivery of drugs *in vitro*. We particularly investigated membrane-

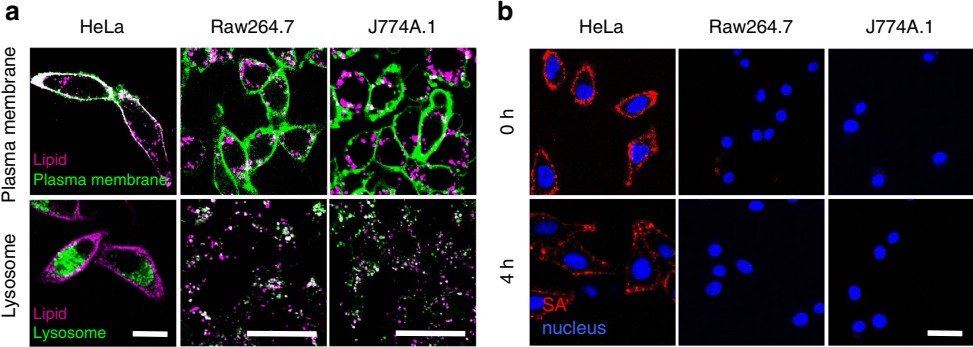

**Figure 3 | Tumour cell membrane-specific delivery of SR-lipids.** (**a**) Confocal fluorescent microscopic images of tumour cells (HeLa) and macrophages (Raw264.7 and J774A.1) after FL-mediated delivery of fluorophore-lipids (magenta). Cells were treated with fluorophore-FLs for 1 h and then imaged. Plasma membranes and lysosomes were stained with CellMask and LysoTracker (green), respectively. (**b**) Confocal fluorescent microscopic images of tumour cells (HeLa) and macrophages (Raw264.7 and J774A.1) treated with fluorophore-SA (red) 0 h (immediately) or 4 h after FL-mediated biotin-lipid delivery. Nuclei were stained with Hoechst (blue). Scale bar represents 20 μm.

targeted phototherapy because membrane photodamage was shown to induce rapid necrosis-like cell death[18,28]. The conjugation of SA to the photosensitizer chlorin e6 (Ce6) resulted in approximately four Ce6 molecules to each SA molecule (Ce6-SA). Cells were treated with biotin-FLs for 1 h, washed, and then incubated with Ce6-SA for 1 or 5 h. Confocal microscopy revealed intrinsic fluorescence of Ce6, primarily on the plasma membrane at 1 h post-incubation, and the majority remained for over 5 h post-incubation (Fig. 4a), indicating the rapid membrane localization of Ce6-SA and its minimal intracellular uptake. For phototherapeutic studies, the cells were irradiated using a 660-nm laser and phototoxicity was measured using the Live/Dead assay. The phototoxicity of membrane-bound Ce6-SA was significantly influenced by the dose of radiation (Fig. 4b and Supplementary Fig. 13), not by the incubation time (Fig. 4c), suggesting that the minimal internalization of Ce6-SA bound to the cell surface preserves the membrane-selective phototherapeutic effects. To explore the SR-dependent membrane targeting of photosensitizers, the cells were incubated with Ce6-SA without biotin-FL pretreatment. Even 5 h of incubation with Ce6-SA did not result in significant cell death after irradiation, implying that Ce6-SA alone neither binds to nor is internalized into cells at a sufficient level to induce phototoxicity (Fig. 4d). In addition, free Ce6-SA in the culture medium, which was not bound to the plasma membrane, did not contribute to phototoxicity (Fig. 4e). We further explored the nature of phototoxicity induced by photosensitizers localized on the cell surface. Phototoxicity experiments were first conducted in the presence of sodium azide, which is a redox quencher and singlet oxygen scavenger, to examine the photodynamic effects of membrane-bound photosensitizers. When the cells were irradiated in the presence of sodium azide, the phototoxicity was significantly reduced and completely cancelled out with 5 mM sodium azide (Fig. 4f). Temperature change of the medium containing Ce6-SA was also measured during irradiation to examine the photothermal effects. No temperature change was observed, even with 500 mM Ce6-SA, which is ten times higher than the concentration used in the phototoxicity experiments (Fig. 4g). From these results, we demonstrated that the irradiation of membrane-bound photosensitizers induces rapid cell death by attacking the cell membrane with reactive oxygen species in close proximity. Collectively, these results illustrate that membrane-specific localization and retention of photosensitizers are crucial for the exertion of substantial phototoxic effects.

We next tested whether membrane-selective targeting and phototoxicity of photosensitizers could be observed on the neighbouring cells via EV-mediated intercellular SR-lipid transfer.

The transwell experiment was performed again, as described in Supplementary Fig. 3. In both the upper filter and the lower chamber, the FL-treated group showed higher phototoxicity than CL- and PL-treated groups (Fig. 4h,i). The phototoxicity of Ce6-SA was scarcely observed in the lower chamber cells when biotin-lipids were delivered to the upper filter cells by CLs and PLs. Thus, we can deduce that membrane-selective delivery of SR-lipids leads to their efficient EV-mediated intercellular transfer and substantial phototoxicity of subsequently membrane-targeted photosensitizers.

Photosensitizer-lipid conjugates could also be delivered to the plasma membranes of cancer cells using FLs and further transferred to the membranes of neighbouring cells via secreted EVs. To examine their potential for membrane-selective phototoxicity, Ce6-conjugated lipids (Ce6-lipids) were prepared and incorporated into the FLs in place of biotin-lipids (Ce6-FLs). Ce6-FLs and biotin-FLs were similar in hydrodynamic size, but the direct conjugation of Ce6 to FLs resulted in a large decrease in the surface charge (Supplementary Fig. 14a). Confocal microscopy and fluorescence quantification revealed that FLs delivered Ce6-lipids mainly to intracellular regions, not onto plasma membranes, and the intracellular quantity was significantly lower than that of Ce6-SA delivered via the cooperative targeting (Supplementary Fig. 14b). Furthermore, FL-mediated Ce6-lipid delivery in the transwell experiment exerted poor cellular phototoxicity in both the upper filter and the lower chamber (Supplementary Fig. 14c). Inefficient membrane fusion and poor phototoxicity of Ce6-FLs are presumably due to a change in the surface characteristics of liposomes, which results from free carboxylic groups in the Ce6 conjugated to the lipid. These findings demonstrate that the cooperative membrane targeting nanosystem is beneficial for localizing a variety of therapeutic agents on the plasma membranes, over a wide range of tissues.

**Cooperative tumour cell membrane-targeted delivery.** Having verified that the cooperative targeting strategy led to substantial membrane localization of targeted agents *in vitro*, we examined the distribution pattern in tumours *in vivo*. First, to observe the tumour distribution of SR-lipids, FLs carrying fluorophore-lipids were intravenously injected into mice bearing mouse breast 4T1 tumours with an average volume of ∼60 mm³, and tumour section images were acquired from 2 to 24 h post-injection. Fluorescent immunohistochemistry of the tumour sections revealed that the fluorophore-lipids accumulating in the perivascular regions spread dramatically towards interstitial tissues over time (Fig. 5a). Importantly, fluorescence was detected throughout the tumour at

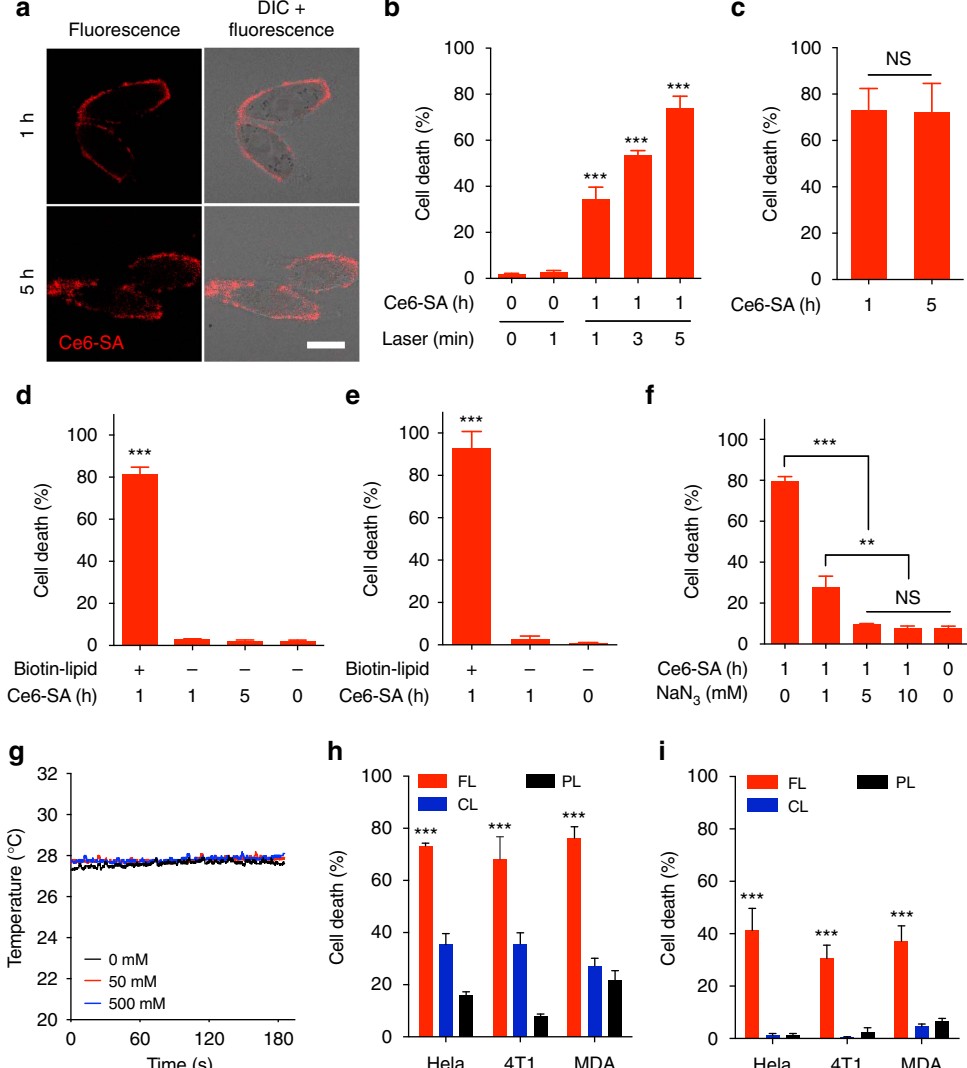

**Figure 4 | Membrane-selective phototoxicity of tumour cells via cooperative membrane-targeted delivery.** (**a**) Confocal fluorescent microscopic images of HeLa cells treated with Ce6-streptavidin (SA) for 1 or 5 h after FL-mediated biotin-lipid delivery. Scale bar represents 20 μm. (**b**) Irradiation dose-dependent cell death after photodynamic therapy (PDT). HeLa cells were treated with Ce6-SA for 1h after FL-mediated biotin-lipid delivery, washed and then irradiated for PDT. (**c**) Incubation time-independent cell death after PDT. HeLa cells were treated with Ce6-SA for 1 or 5 h after FL-mediated biotin-lipid delivery, washed and then irradiated for PDT. (**d**) FL-dependent cell death after PDT. Cells were treated with Ce6-SA for 1 or 5 h directly or after FL-mediated biotin-lipid delivery, washed and then irradiated for PDT. (**e**) Phototoxic effect of free Ce6-SA in the media. Cells were treated with Ce6-SA for 1h directly or after FL-mediated biotin-lipid delivery and then irradiated for PDT without washing. (**f**) Reactive oxygen species-dependent cell death after PDT. Cells were treated with Ce6-SA for 1h after FL-mediated biotin-lipid delivery and washed. They were then treated with sodium azide (NaN$_3$) at different concentrations and irradiated for PDT. (**g**) Temperature change in the medium containing Ce6-SA at different concentrations during irradiation. (**h** and **i**) Phototoxicity of tumour cells [HeLa, 4T1 and MDA-MB-231(MDA)] in the upper transwell filter (**h**) and the lower chamber (**i**) treated with Ce6-SA 4 h after biotin-lipid delivery to cells in the upper filter using various types of liposome. Data are means ± s.e.m. ($n = 6$ for **b**, $n = 5$ for **f**, $n = 3$ for **c–e,h** and **i**; NS, not significant; ***$P < 0.001$, Student's $t$-test for **b–e,h** and **i**; **$P < 0.01$, ***$P < 0.001$, one-way analysis of variance (ANOVA) with Tukey's *post hoc* test for **f**).

24 h post-injection. In contrast, fluorophore-lipids delivered by PLs, which are the most common liposomal formulation for tumour-specific drug delivery[29], accumulated predominantly near blood vessels, even at 24 h post-injection. Uniform distribution of fluorophore-lipids delivered by FLs was also observed in relatively large tumours (~ 360 mm$^3$; Supplementary Fig. 15; Supplementary Methods). To prove that EVs primarily mediate tumour penetration of SR-lipids *in vivo*, mice bearing 4T1 tumours with systemic depletion of EVs were prepared by intraperitoneally injecting DMA daily for 3 days[30]. DMA treatment decreased quantity of blood EVs by 65% (Supplementary Fig. 16a). Systemic administration of fluorophore-FLs into the EV-depleted mice led to significantly reduced penetration of fluorophore-lipids in

tumour tissues, implying that EVs in the tumour microenvironment play an important role in mediating tissue distribution of fluorophore-lipids delivered to the membranes of perivascular tumour cells (Supplementary Fig. 16b). Next, to investigate how the membrane-selective delivery of SR-lipids affects the accumulation and distribution of subsequently administered targeted agents in tumours, fluorophore-SA was intravenously injected into mice 24 h after biotin-FL injection. At 48 h after fluorophore-SA injection, the tumours were excised and subjected to fluorescent immunohistochemistry and accumulation profile analysis. The peripheral regions where blood vessels are relatively permeable and the interstitial regions where blood vessels are often collapsed under high interstitial fluid pressure were

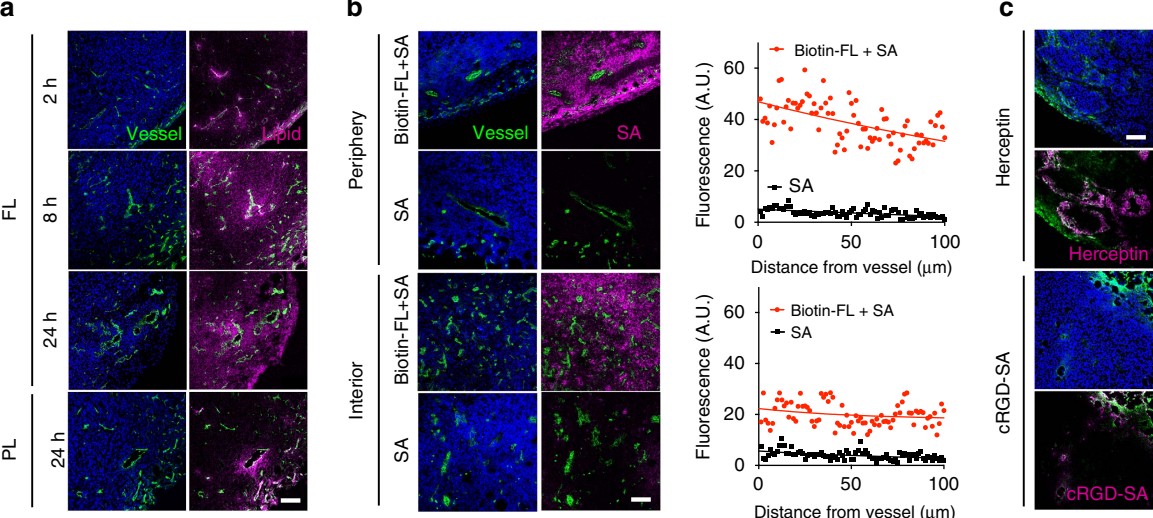

**Figure 5 | Cooperative tumour cell membrane-targeted delivery.** (**a**) Fluorescence images of 4T1 tumour sections after intravenous injection of FL or PL incorporating fluorophore-lipids (magenta). (**b**) Fluorescence images of 4T1 tumour sections and accumulation profiles of fluorophore-streptavidin (SA) after two-step targeted delivery. Mice bearing 4T1 tumours were intravenously injected with biotin-FLs or phosphate-buffered saline 1 day before the intravenous injection of fluorophore-SA (magenta). Tumours were collected 2 days after fluorophore-SA injection. (**c**) Fluorescence images of NCI-N87(Her2 +) and MDA-MB-231(αvβ3 integrin +) tumour sections 48 h after intravenous injection of Herceptin (magenta) and cRGD-SA conjugated with fluorophores (magenta), respectively. Nuclei were stained with Hoechst (blue) and vessels with CD31 (green). Scale bar represents 100 μm.

specifically observed to compare the accumulation and distribution patterns[31]. The cooperative membrane targeting nanosystem exhibited superior accumulation (greater than fivefold) and more uniform distribution of fluorophore-SA in both peripheral and interior regions compared with injection with fluorophore-SA alone (Fig. 5b), which was also hardly observed with conventional tumour targeting systems, Herceptin (trastuzumab, HER2-targeted antibody)[32] and cRGD-SA ($\alpha_v\beta_3$ intergin-targeting peptide-coated protein)[33] (Fig. 5c). In addition, this cooperative targeting did not cause alterations in fluorophore-SA accumulation in the liver and spleen, where biotin-FLs were largely cleared out (Supplementary Fig. 17; Supplementary Methods), presumably due to poor membrane localization of biotin-lipids on macrophages, as shown in Fig. 3a,b. These results illustrate that the membrane localization of SR-lipids occurs preferentially in tumours compared with that in other normal organs, including liver and spleen. Taken together, these observations demonstrate that tumour cell membrane-selective delivery and subsequent EV-mediated intercellular transfer of SR-lipids result in enhanced tumour-specific accumulation and the homogeneous distribution of targeted agents.

**Cooperative tumour cell membrane-targeted phototherapy.** We next examined the phototherapeutic efficacy of the cooperative membrane-targeting nanosystem *in vivo*. 4T1 mouse breast tumours with an average volume of ∼60 mm³ were irradiated with a single dose of light (∼100 mW cm⁻², 660 nm, 30 min) 2 days after Ce6-SA injection alone or cooperative targeting with various types of liposome, and their volumes were measured over 2 weeks. Single treatment with the cooperative membrane-targeting nanosystem (biotin-FL + Ce6-SA) induced a significant reduction in tumour growth after irradiation compared with those in the control, untargeted (Ce6-SA alone), and cooperative membrane-untargeted groups (biotin-CL/PL + Ce6-SA; Fig. 6a). Histological analysis also revealed that the quantity of viable cells significantly decreased throughout the tumour after cooperative membrane-targeted phototherapy (Fig. 6b). We finally evaluated the phototherapeutic efficacy of the

cooperative membrane-targeting nanosystem in mice bearing single MDA-MB-231 human breast tumours. MDA-MB-231 tumour is one of the triple-negative breast cancer types lacking the most commonly targeted receptors (oestrogen receptor, progesterone receptor and human epidermal growth factor receptor 2/neu), which makes the targeted therapies inefficient[34,35]. Thus, pre-decoration of SRs throughout MDA-MB-231 tumours could enhance the efficacy of targeted therapy. Single treatment with the cooperative membrane-targeting nanosystem (biotin-FL + Ce6-SA) resulted in complete tumour regression after irradiation, while no phototherapeutic effects were observed in other treatment groups (Fig. 6c and Supplementary Fig. 18). For all of the treatments evaluated in this work, no significant loss in body mass was observed, indicating their negligible systemic toxicity (Supplementary Fig. 19). Overall, these *in vivo* phototherapeutic studies demonstrate that the cooperative membrane-targeting nanosystem significantly enhanced the phototherapeutic efficacy due to membrane localization of photosensitizers throughout the tumour and membrane-specific amplification of phototoxicity.

**Discussion**

For effective cancer therapy, it is necessary to deliver anti-cancer agents to all of the cells within tumour tissues at a sufficient level to exert a therapeutic effect. This has been challenging since tumour microenvironments do not allow the homogeneous distribution of therapeutic agents, restricting their effects primarily to perivascular regions. Here, we developed a cooperative membrane-targeting nanosystem that exploits biological carriers, EVs, for the generation of SRs on cellular membranes throughout tumour tissues, which can greatly enhance the tumour accumulation and distribution of therapeutics. We demonstrated that the primary delivery of SR-lipids to tumour cell membranes via both FLs and EVs led to substantial membrane-selective targeting of the following photo-sensitizers, thereby dramatically enhancing the phototherapeutic outcome. This therapeutic benefit probably resulted from three components: the uniform tumour distribution of SRs due to

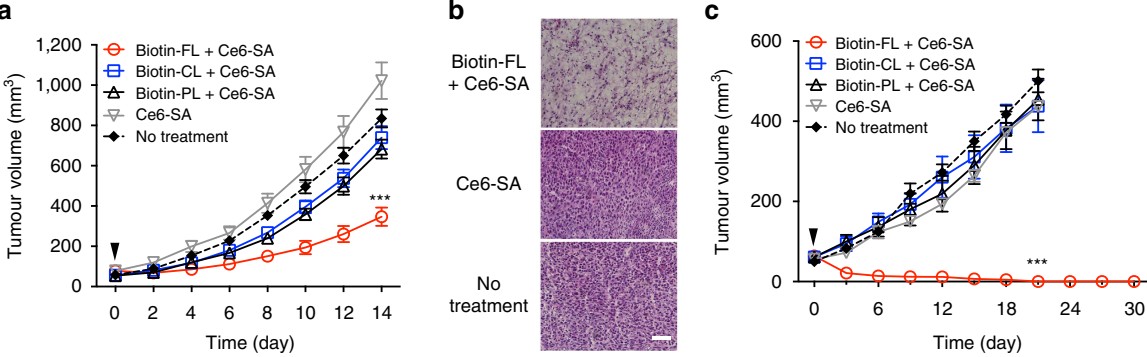

**Figure 6 | Cooperative tumour cell membrane-targeted phototherapy.** (**a**) Tumour growth inhibition by cooperative membrane-targeted phototherapy in 4T1 tumours. (**b**) Histological observation of 4T1 tumours visualized using haematoxylin and eosin staining at day 2 after cooperative membrane-targeted phototherapy. Scale bar represents 100 μm. (**c**) Tumour growth inhibition by cooperative membrane-targeted phototherapy in MDA-MB-231 tumours. Arrowheads indicate laser irradiation after two-step targeted delivery. Data are mean ± s.e.m. (n = 6 in each group, ***P < 0.001 for treatment compared with the other groups using one-way analysis of variance (ANOVA) with Tukey's *post hoc* test).

active EV-mediated lipid transport, specific recognition of the cell surface SRs by the targeted agents and membrane-localized phototherapeutic effects. We believe that this cooperative membrane-targeting approach would significantly improve therapeutic efficacy in tumours that possessed a limited quantity of specific membrane receptors for targeted therapy.

## Methods

**Cells.** HeLa human cervical cancer cells (ATCC CCL-2) and NIH-3T3 fibroblasts (ATCC CRL-1658) were maintained in Dulbecco's Modified Eagle's Medium (Hyclone, South Logan, UT, USA), and 4T1 mouse mammary carcinoma cells (ATCC CRL-2539), MDA-MB-231 human breast cancer cells (ATCC HTB-26), Raw264.7 and J774A.1 mouse macrophage cells (ATCC TIB-71 and ATCC TIB-67), and NCI-N87 human gastric cancer cells (ATCC CRL-5822) were maintained in RPMI medium (Hyclone). The media were supplemented with 10% fetal bovine serum (Hyclone) and 1% penicillin-streptomycin (Hyclone). Human umbilical vein endothelial cells (ATCC CRL-1730) were maintained in endothelial cell growth medium, EGM-2 Bulletkit (Lonza). All cells were cultured in tissue culture flasks in a humidified incubator at 37 °C in an atmosphere of 95% air and 5% carbon dioxide.

**Liposome synthesis.** 1,2-Dimyristoyl-sn-glycero-3-phosphocholine (DMPC; Avanti Polar Lipids, Alabaster, AL, USA), 1,2-distearoyl-sn-glycero-3-phos-phoethanolamine-N-[methoxy(polyethylene glycol)-2000] (DSPE-PEG; Avanti Polar Lipids), 1,2-dioleoyl-3-trimethylammonium-propane (DOTAP; Avanti Polar Lipids) and 1,2-dipalmitoyl-sn-glycero-3-phosphoethanolamine-N-(biotinyl) (biotinylated PE; Avanti Polar Lipids) were used to prepare liposomes. The molar ratios of DMPC, DSPE-PEG, DOTAP and biotinylated PE used for biotin-FLs, biotin-CLs and biotin-PLs were 71.2:3.8:20:5, 75:0:20:5 and 91.2:3.8:0:5, respectively. For Ce6-FLs, the molar ratio of DMPC, DSPE-PEG, DOTAP and Ce6 were 71.2:3.8:20:5. For fluorophore-FLs, fluorescent lipids, 1,2-dipalmitoyl-sn-glycero-3-phosphoethanola-mine-N-(lissamine rhodamine B sulfonyl) (Liss Rhod PE; Avanti Polar Lipids), were added instead of biotinylated PE. For Cy7-FLs, 16:0 Caproylamine (Avanti Polar lipids) was added instead of biotinylated PE, and aminated liposomes were conjugated with Cy7-NHS Ester (GE Healthcare). For liposome preparation, lipids were dissolved in chloroform and then completely dried overnight. The remaining lipid film was hydrated using phosphate-buffered saline and then extruded through 100-nm membrane pores (Whatman, Little Chalfont, UK). The hydrodynamic size and zeta potential of liposomes were measured using dynamic light scattering (Zetasizer Nano ZS90; Malvern Instruments, Malvern, UK).

**Synthesis of streptavidin conjugates.** For Ce6-SA, carboxylic groups of Ce6 (Santa Cruz Biotechnology, Santa Cruz, CA, USA) were activated by reacting Ce6 dissolved in anhydrous dimethyl sulfoxide (DMSO; 10 mg ml⁻¹) with equimolar N-hydroxysuccinimide (Sigma-Aldrich, St Louis, MO, USA) and 1-ethyl-3-(3-dimethylaminopropyl) carbodiimide (Sigma-Aldrich) dissolved in DMSO (10 mg ml⁻¹) for 1 h at room temperature. The DMSO solution was then mixed with SA (Pierce Biotechnology, Rockford, IL, USA) dissolved in phosphate-buffered saline (5 mg ml⁻¹) for 2 h at room temperature. The unbound Ce6 molecules were removed using PD-10 desalting columns (GE Healthcare, Little Chalfont, UK). For fluorophore-SA, SA dissolved in phosphate-buffered saline (5 mg ml⁻¹) was mixed with a 10-molar excess of Alexa Fluor 594 succimidyl esters (Thermo Fisher Scientific, Waltham, MA, USA). dissolved in DMSO (5 mg ml⁻¹) for 2 h at room temperature. The unbound fluorophores were removed using PD-10 desalting columns. The number of Ce6 or fluorophore molecules conjugated to

each streptavidin molecule was approximately two to four throughout the experiments. The hydrodynamic size and zeta potential of Ce6-SA were measured using dynamic light scattering (Zetasizer Nano ZS90; Malvern Instruments).

*In vitro* **fluorescence cellular imaging.** To observe biotin-lipid delivery to plasma membranes by each type of liposome, we treated HeLa cells with medium containing 140 μM of each liposome for 1 h at 37 °C. After the thorough washing of free lipo-somes, cells were treated with medium containing 50 μM fluorophore-SA to detect biotin on the cell surface. A transwell assay was used to observe EV-mediated intercellular lipid transfer. We treated HeLa cells in the upper filter with 400-nm pores with each type of biotin-liposome for 1 h and co-incubated them with fresh cells in the lower chamber for 4 h. The cells in the lower chamber were incubated with fluorophore-SA for 1 h to detect surface biotins. Surface biotin was quantified by measuring the fluorescence intensity of fluorophore-SA on the cell surface by confocal microscopy at various time points (0, 4, 12, 24 and 48 h). To observe tumour cell-selective SR-lipid delivery by liposomes, we treated tumour cells (HeLa, 4T1 and MDA-MB-231), macrophages (Raw264.7 and J774A.1) or non-parenchymal cells (human umbilical vein endothelial cell and NIH-3T3) with fluorophore-FLs for 1 h. Cells were also stained with LysoTracker (Thermo Fisher Scientific) or Vybrant DiI Cell-Labeling solution (Thermo Fisher Scientific) to visualize lysosomes or plasma membranes, respectively, and imaged using a confocal microscope (Nikon). The binding of fluorophore-SA on the cells was observed after treatment of the cells with biotin-FLs. The cells were incubated with fluorophore-SA immediately, and 4 h after biotin-FL treatment and imaged using the confocal microscope. All experiments in this section were repeated at least three times and they showed similar results.

**EV isolation and quantification.** For EV-depleted supernatant treatments, HeLa cells were treated with biotin-FLs for 1 h, washed to remove free liposomes in the media and further incubated for 4 h. Dead cells and debris were first removed from the supernatant of the treated cells by centrifugations at 10,000g for 30 min at 4 °C. Then EVs were removed or isolated from the supernatant by ultracentrifugation at 100,000g for 120 min at 4 °C. Fresh cancer cells were treated with EV-containing or EV-depleted supernatant for 1 h, incubated with fluorophore-SA for 1 h and imaged using confocal microscopy. To block the EV secretion pathway, HeLa cells in the transwell filter were pretreated with 300 μl of the transfection solution containing 25 pmole ARF6 (Invitrogen, Catalogue number 4390824) or Rab27 siRNA (Invitrogen, Catalogue number 4390824) and 0.5 μl LipofectamineTM (Invitrogen) 1day before they were treated with biotin-FLs. The cells were then co-incubated with fresh cells in the lower chamber for 4 h. The cells in the lower chamber were incubated with fluorophore-SA for 1 h, and imaged using confocal microscopy. For chemical inhibitor-induced depletion of EVs, HeLa cells were incubated for 4 h in the presence of 0 or 200 μM DMA (inhibitor of exosome secretion[27], Abcam). Then, the cells were treated with 140 μM FLs containing fluorophore-lipids for 1 h, washed thoroughly to remove free liposomes in the media, and further incubated for 24 h. The secreted EVs were collected using the ultracentrifugation protocol and their fluorescence was measured using a spectrofluorometer (Gemini XPS; Molecular Devices). For direct observation of EV incorporation of SR-lipids, HeLa cells were treated with 140 μM liposomes containing fluorophore-lipids for 1 h, washed thoroughly to remove free liposomes in the media, and further incubated for 24 h. The secreted EVs were collected using the ultracentrifugation protocol and diluted to obtain between 10 and 100 particles per image. Nanoparticle tracking analysis measurements were performed to quantify fluorescent EVs using a NanoSight NS300 (Malvern). All experiments in this section were repeated at least three times and they showed similar results.

**In vitro phototoxicity.** In all in vitro phototoxicity experiments, appropriate numbers of cells were seeded into 48-well plates and incubated at 37 °C for 24 h. Cells were treated with liposomes and Ce6-SA for 1 h at 150 and 50 μM, respectively, and washed thoroughly before the next step, unless otherwise mentioned. Laser power density and irradiation time were fixed to 100 mW cm$^{-2}$ and 3 min, respectively, unless otherwise mentioned. To observe irradiation dose-dependent cell death, HeLa, 4T1 and MDA-MB-231 cells were treated with medium containing biotin-FLs for 1 h at 37 °C. Then, the cells were treated with medium containing Ce6-SA for 1 h at 37 °C and irradiated with a laser for 1, 3 or 5 min. After irradiation, phototoxicity was compared with that in the control group. For the phototoxicity assay, the Live/Dead assay was used. After irradiation, cells were treated with 5 μM ethidium homodimer-1 (Invitrogen) at room temperature for 1 h. Red fluorescence from cells with damaged membranes was observed using a fluorescence microscope (Nikon). To observe the internalization of Ce6-SA bound to the cell surface, we treated HeLa cells with biotin-FLs for 1 h and Ce6-SA was applied for 1 or 5 h. To observe biotin-dependent membrane targeting of Ce6-SA, HeLa cells were treated with Ce6-SA for 1 and 5 h without biotin-FL pretreatment. Phototoxicity was measured and compared with that in the control group treated with both biotin-FLs and Ce6-SA. To observe the phototoxicity of Ce6-SA in the medium, HeLa cells treated with and without biotin-FLs were incubated in medium containing Ce6-SA for 1 h and then irradiated. To observe reactive oxygen species-dependent cell death, HeLa cells were treated with biotin-FLs, washed, and then treated with Ce6-SA. After the washing off of free Ce6-SA, the cells were irradiated in medium containing 0, 1, 5 and 10 mM sodium azide. To observe the phototoxicity of HeLa, 4T1 and MDA-MB-231 cells in the upper transwell filter, the cells were treated with liposomes to deliver biotin-lipids on the cell membranes. Then, the cells were treated with Ce6-SA and irradiated. Cells in the lower chamber were co-incubated with an upper transwell filter treated with each type of liposome for 4 h. Then, the cells were treated with Ce6-SA and irradiated. All experiments in this section were repeated at least three times and they showed similar results.

**Half life in blood.** All studies in mice were approved by the Korea Advanced Institute of Science and Technology (KAIST) Committee on Animal Care. FLs and fluorophore-SA were prepared as described above. 5-week-old female Balb/c mice were injected through the tail vein with 200 μl of FL or fluorophore-SA solution. At five different time points after injection, blood was collected by retro-orbital eye bleeding. The fluorescence of blood samples was measured and the half life of each material in blood was determined by fitting blood fluorescence to a single-exponential equation for a one-compartment open pharmacokinetic model. All experiments in this section were repeated at least three times and they showed similar results.

**In vivo fluorescence tissue imaging.** To observe SR-lipid delivery in tumour tissues, tumour models were generated by implanting $5 \times 10^5$ 4T1 cells in 5-week-old female Balb/c mice. When the tumour volume reached ∼60 or 360 mm$^3$, mice were injected with 200 μl of 5 μM of either FLs or PLs containing fluorescent lipids. After 24 h, the mice were killed for histological analysis. Tissue samples from the killed mice were promptly immersed in liquid nitrogen and stored at −80 °C. The tissues were sectioned (10 μm), brought up to room temperature, air-dried for 1 h, and fixed in either acetone or 4% formaldehyde. For vessel staining, fixed sections were treated with blocking solution for 20 min before treatment with rat anti-mouse CD31 (Invitrogen, Catalogue number RM5200) diluted in blocking solution. At 2 h, the tissue sections were washed thoroughly and treated with Alexa Fluor 488-conjugated goat anti-rat IgG (Invitrogen, Catalogue number A-11006) diluted in blocking solution for 1 h. After thorough washing, the tissue sections were mounted in mounting medium (Sigma-Aldrich). The tissue sections were then examined under a confocal microscope (Nikon). The distribution of SR-lipids in tumour tissues was studied by observing fluorophore-lipids using a confocal microscope and NIS-elements BR software (Nikon). To observe the SA accumulation in tumour tissues, mice implanted with a 4T1 tumour were injected with 200 μl of 5 μM of either biotin-FLs or phosphate-buffered saline. After 24 h, the injection of 500 μg of fluorophore-SA was performed. At 2 days, the mice were killed for histological analysis. The distribution of SA in tumour tissues was studied by observing fluorophore-SA using a confocal microscope and NIS-elements BR software. For the quantification of fluorophore-SA accumulation (accumulation profile), an arbitrary line perpendicular to a blood vessel was drawn and fluorescence along the line was measured. Each dot represents the average fluorescence of SA at the distance. We examined 20 vessels from at least three sections from each tumour sample (n = 5). Images of tumour interior regions were taken at least 2 mm away from the edge of tumour. For systemic depletion of exosomes, mice bearing 4T1 tumours were intraperitoneally injected with PBS or 10 μmol kg$^{-1}$ DMA daily for 3 days[30]. On the next day, EVs were isolated from mouse blood, and their amount was measured by the activity of acetylcholinesterase. To examine the effect of systemic exosome depletion on tumour penetration of SR-lipids, EV-depleted mice were intravenously injected with 200 μl of 5 μM of FLs containing fluorescent lipids. After 24 h, tumour samples were collected and the distribution of fluorescent lipids was observed by confocal microscopy. To observe tumour distribution of conventional targeting agents, HER2-positive human gastric tumour and human breast tumour models were generated by implanting $5 \times 10^6$ NCI-N87 and MDA-MB-231 in 5-week-old female Balb/c nude mice, respectively. For fluorescent cRGD-conjugated

streptavidin, streptavidin was incubated with 10-molar excess of Alexa Fluor 594 for 2 h and free dyes were removed using PD-10 desalting columns. Then, SA conjugated with Alexa Fluor 594 was reacted with 10-molar excess of sulfo-SMCC (Sigma-Aldrich) for 30 min, washed on the PD-10 column and incubated with 10-molar excess of cRGD for 30 min. Free cRGD was washed using PD-10 columns. Trastuzumab was conjugated with Alexa Fluor 594 succinimidyl esters according to manufacture's protocol (Thermo Fischer Scientific). When tumour volumes reached ∼50–70 mm$^3$, mice bearing NCI-N87 tumours were injected with 500 μg of trastuzumab (Herceptin) conjugated with Alexa Fluor 594 succinimidyl esters and mice bearing MDA-MB-231 tumours with 1.5 mg of streptavidin conjugated with cRGD and Alexa Fluor 594 succinimidyl esters. After 2 days, tumour samples were collected and the distribution of fluorescent targeting agents was observed by confocal microscopy. All experiments in this section were repeated at least three times and they showed similar results.

**In vivo phototherapy.** Tumour models were generated by implanting $5 \times 10^5$ 4T1 cells into 5-week-old female Balb/c mice and $1 \times 10^6$ MDA-MB-231 cells into 5-week-old female Balb/c nude mice. When the tumour volume reached around ∼60 mm$^3$, the mice were randomized into five groups for the following treatments (n = 6): (i) The mice were injected with 200 μl of 10 mM biotin-FLs 1 day before the injection of Ce6-SA (1.5 mg kg$^{-1}$ Ce6). At 2 days, the tumours were irradiated (660 nm, ∼100 mW cm$^{-2}$, 30 min). (ii) The mice were injected with 200 μl of 10 mM biotin-CLs 1 day before the injection of Ce6-SA. At 2 days, the tumours were irradiated. (iii) The mice were injected with 200 μl of 10 mM biotin-PLs 1 day before the injection of Ce6-SA. At 2 days, the tumours were irradiated. (iv) The mice were injected with Ce6-SA without biotin-FL pre-treatment. At 2 days, the tumours were irradiated. (v) No treatment was performed. Tumour volumes were measured at 2–3 day intervals by a blinded investigator with calipers using the following equation: tumour volume $= L \times W^2/2$, where $L$ indicates the length of the long side and $W$ indicates that of the short side. For histological analyses, tissue samples from killed mice were harvested 48 h post-treatment, promptly immersed in liquid nitrogen and stored at −80 °C. The tissues were sectioned (10 μm), brought up to room temperature, air-dried for 1 h and fixed in 4% formaldehyde. The fixed tumour sections were stained following the hematoxylin and eosin staining protocol (Sigma-Aldrich). All experiments in this section were repeated at least three times and they showed similar results.

**Statistical analysis.** Data are presented as mean ± s.e.m. Statistical differences were calculated with an unpaired two-tailed Student's t-test or one-way analysis of variance followed by Tukey's multiple comparison post test using GraphPad Prism 5.0 (GraphPad software). P values $< 0.05$ were considered significant (*$P < 0.05$, **$P < 0.01$ and ***$P < 0.001$). Variances were similar between groups, as determined by the F test using GraphPad Prism 5.0. No statistical method was used to pre-determine sample size. For in vivo phototherapy studies, mice were assigned randomly to treatment groups. The investigators were blinded when assessing the tumour size with calipers. There were no mice excluded from the experiments for statistical analysis. All experiments were routinely repeated at least twice and representative data are reported.

**Data availability.** The data that support the findings of this study are available from the corresponding author on reasonable request.

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

## Acknowledgements

This work was supported by the Basic Science Research Program (Grant No. NRF-2015R1A1A05001420) through the National Research Foundation funded by the Ministry of Science, ICT & Future Planning, and the National R&D Program for Cancer Control (Grant No. 1220070) funded by the Ministry for Health and Welfare, Republic of Korea.

## Author contributions

H.K. and J.-H.P. conceived and designed the research. H.K., J.L. and C.O. carried out the experiments. H.K., J.L., C.O. and J.-H.P. analysed the data. H.K. and J.-H.P. wrote the manuscript.
