## [Peer Review File · Nature Communications]

Reviewers' comments:

Reviewer #1 (Remarks to the Author):

The submitted manuscript by Heegon Kim reports on a cooperative targeting system in which synthetic and biological nano-components participate together in the tumor cell membrane-selective localization of synthetic receptor-lipid conjugates (SR-lipids) to amplify the subsequent targeting of therapeutics through extracellular vesicles. Overall the authors have shown the feasibility of this approach and the mechanism behind it in *in vitro* and *in vivo* studies. However, the wide applicability of this approach may be hindered by body-wide toxic side effects. The authors have touched upon this, however more experiments would be beneficial for the article. Therefore I have a few concerns that need to be addressed before accepting the work for publication. Some of the concerns raised are, in my opinion, mandatory to address (see below).

Major concerns:

1. Authors have used various techniques to inhibit EV secretion in order to prove the involvement of EVs in the transfer of SR, however there is no data depicting the effect of these inhibitors on EV secretion. Therefore, authors should quantify the number of EVs after treating the cells with various inhibitors with for example NTA analysis and include it as supplementary data.
2. Authors should also characterise the EVs from engineered and wildtype cells, by western blot looking for various exosomal markers and electron microscopy in order to show if the treatment of the cells with FL affects the morphology and physiology of EVs.
3. Authors should also perform immuno-EM with immuno gold streptavidin, in order to show that SR are on the surface of EVs. This is a very important experiment to confirm EV incorporation of the SR.
4. Another important experiment to show that it is in fact loaded onto/into EVs is to perform a sucrose or Optiprep gradient experiment and show that the FL-biotin lipid is in fact distributing to the EV compartment of the sucrose gradient. This is "must do" experiment; simply measure the fluorescence in each fraction.
5. It would be interesting for the readers to know how much of the EVs that have incorporated FLs. I do not think this can be assessed by FACS, since FACS cannot measure particles under 150 nm. This can be done by measuring number of fluorescent particles in an NTA or perform FCS or similar techniques.
6. One interesting experiment, which would further prove that this effect is EV mediated is by adding Heparin for blocking EV uptake in the recipient cells and observe the efficiency of transfer. The blockers used in the manuscript are good, however this would add extra weight.
7. The authors have shown very well through experiments how FLs are inefficient in macrophages, endothelial and fibroblasts cells, however authors missed a key experiment to see how well tumour cells can transfer these SR to macrophages, endothelial and fibroblasts through EVs. Also can the authors please give a more detailed explanation why the FLs are not incorporated in the membranes of these cells. Despite the fact that they are endocytosed they should in theory fuse with the membrane as well.
8. Can the authors please also check for FL-EVs in serum at different time points and not only analyse the half-life on FLs.
9. Authors claim through *in vitro* experiments that SR lipids are not membrane localised preferentially in liver and spleen, however authors cannot correlate *in vitro* observations with that of *in vivo*. Can the authors please explain the findings in supplementary figure 14? To me it seems that these SR receptors are primarily enriched in kidney and tumour followed by liver and spleen. Is there a risk of kidney and/or liver toxicity? Is it possible for the authors to exchange the phototoxicity molecule to a toxic molecule induced by a small molecule instead and investigate the body wide effect of the treatment? Just to make sure the specificity of the treatment towards tumour tissue. This experiment is very important to perform in one way or the other to inform the readers of the applicability of the technique. Both kidney and liver toxicity is a major problem in a clinical setting and needs to be further evaluated.

Minor concerns:

1. Author has elucidated the involvement of EVs in the transport of SR lipids very well, however most of the information are in supplementary figures, therefore author can add some of the key data in a new figure in the manuscript.

Reviewer #2 (Remarks to the Author):

Interesting paper, but complicated story.

Extracellular vesicles (EVs) are known to mediate intercellular communications and also known to play a supportive role in tumor progression. EVs are essentially made from cell membranes by cells. The authors claim that the formation EVs can be useful in drug delivery. EVs can travel along the tumor tissue and they can reach into the "deep" tumor area. For large surface modified nanoparticles it has been challenging to go deep into the tumor tissue.

In the new technique they modify the cellular membrane by using fusogenic liposomes (FL). FL basically fuse into the plasma membrane of the cell. Using this technique, they can essentially cause cells to express biotin on their surfaces. Then, they use Ce6 conjugated Streptavidin (Ce6-SA) to target the tumor. They have shown that their system can reach deep into tumors and get better results with such PDT treatment.

Comments

- The work presents an interesting idea and is novel.
- A full paper rather than a letter would be preferable. A section by section demonstration of the work can guide the readers to follow the manuscript better.
- Because of the need to utilize fusogenic liposome (FL) before the PDT treatment, it means it needs an additional step prior to the PDT treatment. How is the fusogenic liposome targeted to only cancer cells? The authors have not clearly demonstrated good, cancer selective, targetability of the FL.
- v Is it possible that FL fuses into other normal epithelial cells?
- Can the EVs travel outside of the tumor tissue?
- The FL will most likely be toxic at higher concentration. Only when tumors are large (>1 cc), does the tumor microenvironment form and thus deep tumor targeting becomes important, while the authors studied an only 50 mm³ sized tumor. As there may be only limited amounts of FL that can be introduced into the animal, would that work for the large tumors for which this new treatment is envisioned. Although theoretically it would be possible for the FL to modify the entire tumor, the volume of the tumor increases at a much faster ratio than the surface area of the tumor. Do the authors believe that enough EVs can be created to cover a bigger tumor volume?
- The comparison with conventional targeted therapy (supplementary figure 17) is an important piece of data which should be included into the main figure.

Reply to reviewers' comments

We have addressed the reviewers' comments as follows:

Reviewer #1 (Remarks to the Author):

1. Authors have used various techniques to inhibit EV secretion in order to prove the involvement of EVs in the transfer of SR, however there is no data depicting the effect of these inhibitors on EV secretion. Therefore, authors should quantify the number of EVs after treating the cells with various inhibitors with for example NTA analysis and include it as supplementary data.

Thank you for your valuable comment. As suggested, we have quantified the number of EVs produced from the cells treated with chemical inhibitor by nanoparticle tracking analysis (NTA) method. As a result, we found that the number of EVs from DMA-treated cells decreased by 60 % compared to that of EVs from PBS-treated cells. We have included the data in **supplementary figure 5b**.

In supplementary figure 5b,

(b) Concentration of EVs produced from DMA-treated cells. HeLa cells were incubated for 4 h in the absence or presence of 200 μM DMA. After 24 h, the secreted EVs were collected, and the concentration was measured by nanoparticle tracking analysis.

In supplementary methods,

“For quantification of EV concentration, the EVs produced from DMA-treated cells were collected using the ultracentrifugation protocol and diluted to obtain between 10 and 100 particles per image. Nanoparticle tracking analysis (NTA) measurements were performed to quantify the concentration of secreted EVs was using a NanoSight NS300 (Malvern).”

2. Authors should also characterise the EVs from engineered and wildtype cells, by western blot looking for various exosomal markers and electron microscopy in order to show if the treatment of the cells with FL affects the morphology and physiology of EVs.

Thank you for your valuable comment. As suggested, we have characterized EVs produced from PBS-treated cells and the FL-treated cells with dynamic light scattering (DLS), transmission electron microscopy (TEM) and western blot. As a result, the incorporation of functional lipids was unlikely to alter morphology, protein profile (including exosomal markers CD63 and CD9), and size of EVs. We have included the data in **supplementary figure 8**.

In main text,

“Importantly, the incorporation of functional lipids was unlikely to alter morphology, protein profile (including exosomal markers CD63 and CD9), and size of EVs (**Supplementary Fig. 8**).”

In supplementary figure 8,

Supplementary Figure 8. Morphology, protein profile and size of EVs after biotin-FL treatment. (a) Representative TEM images of EVs from cells treated with either PBS (control) or biotin-FLs. (b) Coomassie blue staining of protein extracts from EVs from cells

treated with either PBS (control) or biotin-FLs. (c) Western blot analysis of exosomal marker proteins CD63 and CD9 on the EVs from cells treated with either PBS (control) or biotin-FLs. (d) Hydrodynamic size of EVs from cells treated with either PBS (control) or biotin-FLs. Scale bar indicates 100 nm. Data are means \pm s.e.m. (n = 3; NS, not significant; Student's *t* test).

In supplementary methods,

“Size measurement

EVs produced from fluorophore/biotin-FL-treated cells were isolated using the ultracentrifugation protocol and resuspended in PBS. The hydrodynamic size of EVs was measured using dynamic light scattering (Zetasizer Nano ZS90; Malvern Instruments, Malvern, UK).

Transmission electron microscopy

The EVs produced from biotin-FL treated HeLa cells were fixed in 2% paraformaldehyde (Sigma-Aldrich) and stored at 4°C before use. 5 μ L of resuspended EVs was deposited on Formvar-carbon coated EM grids (Ted Pella, Inc.) for 20 min until it dries. For nanogold-staining, the grids were blocked using blocking solution (1% BSA, 5% goat serum, and 0.02% Tween) for 10 min, and incubated with streptavidin-nanogold (1.4 nm, 1/50 diluted, Nanoprobe, Catalog number 2016) for 30 min. The grids were then washed with PBS for 6 times. For stabilization, the grids were transferred to a 50 μ L drop of 1% glutaraldehyde for 5 min before transferring to a 100 μ L drop of distilled water for 2 min. This was repeated 7 times for a total of 8 water washes. The grids were kept wet on the side of the membrane during all steps, but dry on the opposite side. For negative staining, the grids were placed onto a 50 μ L drop of 2% phosphotungstic acid (Sigma-Aldrich) for 2 min. Transmission electron microscopic images were obtained using a JEM-2100F HRTEM operating at 200 kV (JEOL).

Western blot

The EVs produced from biotin-FL-treated cells were isolated using the ultracentrifugation protocol and resuspended in PBS. The protein concentration of EVs was measured by performing BCA assay. The EVs were concentrated using a 100K Amicon centrifugal filter (Millipore, Billerica, MA, United States) to obtain 2 mg/ml protein concentration. 30 μ L of EV solution was mixed with 10 μ L of 4x Laemmli buffer (Bio-Rad) and boiled for 10 min at 95°C. Proteins were resolved by SDS-PAGE, transferred to polyvinylidene fluoride (PVDF) membrane. The protein profile of EVs was visualized on the PVDF membrane using Coomassie blue staining according to the manufacturer's protocol (Bio-rad). For immunoblotting, the PVDF membrane was blocked in 5% skim milk for 1 h. The membranes were treated with CD63 antibody (Santa Cruz Biotechnology, Catalog number sc-15363) or CD9 antibody (Santa Cruz Biotechnology, Catalog number sc-9148), and incubated overnight at 4°C. The membranes were washed 3 times with Tris-buffered saline with 0.1% Tween 20 (TBST) and treated with Horseradish Peroxidase-linked secondary antibody (Santa Cruz Biotechnology, Catalog number sc-2030) for 1 h. The membranes were washed 3 times with TBST. Protein bands were detected using X-ray film and enhanced using chemiluminescence reagent (Bio-Rad, Catalog number 1705061).”

3. Authors should also perform immuno-EM with immno gold streapdividn, in order to show that SR are on the surface of EVs. This is a very important experiment to confirm EV incorporation of the SR.

Thank you for your valuable comment. As suggested, we have performed immune-EM on EVs produced from cells treated with either PBS (control) or biotin-FLs with streptavidin-nanogold(1.4 nm) conjugate. As a result, EVs from biotin-FL-treated cells showed binding of gold nanoparticles on the surface while EVs from PBS-treated cells did not. These results indicate that the surface of EVs produced from biotin-FL-treated cells is decorated with SRs, and provides binding sites for targeted agents. We have included the data in **supplementary figure 7**.

In main text,

“The presence of functional lipids in the EVs was further confirmed with the results of sucrose gradient ultracentrifugation and transmission electron microscopy (**Supplementary Figs. 6 and 7**).”

In supplementary figure 7,

Supplementary Figure 7. Representative TEM images of unmodified EVs and biotin-EVs after streptavidin-nanogold staining. EVs from the cells treated with either PBS (control) or biotin-FLs were stained with streptavidin-nanogold(1.4 nm). EVs from biotin-FL-treated cells showed binding of gold nanoparticles on the surface while EVs from PBS-treated cells did not. Scale bar indicates 100 nm.

In supplementary methods,

“Transmission electron microscopy

The EVs produced from biotin-FL treated HeLa cells were fixed in 2% paraformaldehyde (Sigma-Aldrich) and stored at 4°C before use. 5 µl of resuspended EVs was deposited on Formvar-carbon coated EM grids (Ted Pella, Inc.) for 20 min until it dries. For nanogold-staining, the grids were blocked using blocking solution (1% BSA, 5% goat serum, and 0.02% Tween) for 10 min, and incubated with streptavidin-nanogold (1.4 nm, 1/50 diluted,

Nanoprobe, Catalog number 2016) for 30 min. The grids were then washed with PBS for 6 times. For stabilization, the grids were transferred to a 50 μ L drop of 1% glutaraldehyde for 5 min before transferring to a 100 μ L drop of distilled water for 2 min. This was repeated 7 times for a total of 8 water washes. The grids were kept wet on the side of the membrane during all steps, but dry on the opposite side. For negative staining, the grids were placed onto a 50 μ l drop of 2 % phosphotungstic acid (Sigma-Aldrich) for 2 min. Transmission electron microscopic images were obtained using a JEM-2100F HRTEM operating at 200 kV (JEOL).”

4. Another important experiment to show that it is in fact loaded onto/into EVs is to perform a sucrose or Optiprep gradient experiment and show that the FL-biotin lipid is in fact distributing to the EV compartment of the sucrose gradient. This is “must do” experiment; simply measure the fluorescence in each fraction.

Thank you for your valuable comment. As suggested, we have performed sucrose gradient ultracentrifugation to verify incorporation of SR-lipids in the vesicles located in the EV compartment of the sucrose gradient. As a result, EVs were enriched in the third fraction which consists of 5th and 6th layer with density of 1.146 and 1.176 g/ml, respectively, which was in agreement with the previous literature [Witwer, K. W. *et al.* Standardization of sample collection, isolation and analysis methods in extracellular vesicle research. *J. Extracell. Vesicles* **2**, 20360 (2013)]. We have included the data in **supplementary figure 6**.

In main text,

“The presence of functional lipids in the EVs was further confirmed with the results of sucrose gradient ultracentrifugation and transmission electron microscopy (**Supplementary Figs. 6 and 7**).”

In supplementary figure 6,

a**b****c**
Supplementary Figure 6. EV purification using sucrose gradient ultracentrifugation. (a) Sucrose gradient and density table. After ultracentrifugation, EVs were enriched in the third fraction which consists of 5th and 6th layer with density of 1.146 and 1.176 g/ml, respectively, which was in agreement with the previous literature². (b) Fluorescence intensity of each collected fraction. (c) Representative size distribution of EVs in the third fraction. Data are means \pm s.e.m. (n = 3).

In supplementary methods,

“Sucrose gradient ultracentrifugation

EV pellet was prepared from 200 ml of cell culture medium from fluorophore-FL-treated HeLa cells using the ultracentrifugation protocol and resuspended in 100 μ l of PBS. EV pellet was resuspended in 100 μ l of PBS. For EV purification, 10 - 90% (10, 16, 22, 28, 34, 40, 46, 52, 58, 64, 70, and 90%) sucrose stocks were prepared with PBS. The EV solution was mixed with 1 ml of 90% sucrose stock solution (final sucrose concentration = 82%) and transferred into 13.2 ml ultra-clear Beckman Ultracentrifuge tubes. Gradient was overlaid slowly on top of the EV solution starting with 1 ml of 70% sucrose solution (from the highest to the lowest sucrose concentration). The end of the pipette tip was in contact with the inside wall of the ultracentrifuge tube and the solution was slowly added into the tube. Ultracentrifugation was performed at 4°C (100,000g for 16 h). After ultracentrifugation, 2 ml fractions from the top to bottom were collected. Each fraction was examined for fluorescence and hydrodynamic size of EVs.”

5. It would be interesting for the readers to know how much of the EVs that have incorporated FLs. I do not think this can be assessed by FACS, since FACS cannot measure particles under 150 nm. This can be done by measuring number of fluorescent particles in an NTA or perform FCS or similar techniques.

Thank you for your valuable comment. As suggested, we have performed NTA to quantify the percentage of EVs packaging fluorophore-lipids out of total EVs produced from the cells treated with liposomes containing fluorophore-lipids. As a result, FL-treated cells produced EVs packaging fluorophore-lipids (>40 %) more efficiently compared with CL- and PL-treated cells (4.4% and 1.7%, respectively). We have included the data in **figure 2h**.

In main text,

“We also quantified the percentage of EVs packaging fluorophore-lipids out of total EVs produced from the cells treated with liposomes containing fluorophore-lipids. FL-treated cells produced EVs packaging fluorophore-lipids (>40 %) more efficiently compared with CL- and PL-treated cells (4.4% and 1.7%, respectively) (Fig. 2h).”

In figure 2h,

(h) Nanoparticle tracking analysis of EVs produced from the cells treated with liposomes containing fluorophore-lipids. Data are means \pm s.e.m. [n = 3, ***p < 0.001, one-way ANOVA with Tukey's post hoc test].

In methods,

“For direct observation of EV incorporation of functional lipids, HeLa cells were treated with 140 μ M liposomes containing fluorophore-lipids for 1 h, washed thoroughly to remove free liposomes in the media, and further incubated for 24 h. The secreted EVs were collected using the ultracentrifugation protocol and diluted to obtain between 10 and 100 particles per image. Nanoparticle tracking analysis (NTA) measurements were performed to quantify fluorescent EVs using a NanoSight NS300 (Malvern).”

6. One interesting experiment, which would further prove that this effect is EV mediated is by adding Heparin for blocking EV uptake in the recipient cells and observe the efficiency of transfer. The blockers used in the manuscript are good, however this would add extra weight.

Thank you for your valuable comment. As suggested, we have used heparin, a known inhibitor of cellular uptake of EVs, to verify EV-mediated intracellular transfer of SR-lipids. As a result, the fluorescence signal of EV-treated cells was significantly reduced in the presence of heparin, indicating that cellular delivery of fluorophore-lipids is mediated by EVs. We have included the data in **supplementary figure 4**.

In main text,

“EV-mediated delivery of functional lipids to neighboring cells was also confirmed with the result showing that cellular uptake of EVs packaging fluorophore-lipids was significantly reduced in the presence of heparin, a known inhibitor of cellular uptake of EVs²⁶ (**Supplementary Fig. 4**).”

In supplementary figure 4,

Supplementary Figure 4. Heparin treatment for inhibition of cellular uptake of EVs. Confocal fluorescent microscopic images and fluorescence quantification of HeLa cells treated with EVs containing fluorophore-lipids in the presence of heparin. EVs from cells treated with FLs containing fluorophore-lipids were isolated from cell culture medium using ultracentrifugation method. Fresh cells were treated with EVs containing fluorophore-lipids in the absence (control) or presence of 100 µg/ml heparin for 2 h, and then imaged with confocal microscopy. Data are means ± s.e.m. (n = 20, ***p<0.001, Student's *t* test). Scale bar represents 20 µm.

In supplementary methods,

“Heparin treatment

HeLa cells were treated with 140 µM FLs containing fluorophore-lipids for 1 h, washed thoroughly to remove free liposomes in the media, and further incubated for 24 h. The secreted EVs were isolated from the supernatant using the ultracentrifugation protocol and resuspended in PBS. Fresh HeLa cells were treated with 100 µg/ml EVs containing fluorophore-lipids in the absence or presence of 100 µg/ml heparin (sigma) for 2 h, and then imaged with confocal microscopy (Nikon, Tokyo, Japan).”

7. The authors have shown very well through experiments how FLs are inefficient in macrophages, endothelial and fibroblasts cells, however authors missed a key experiment to see how well tumour cells can transfer these SR to macrophages, endothelial and fibroblasts through EVs. Also can the authors please give a more

detailed explanation why the FLs are not incorporated in the membranes of these cells. Despite the fact that they are endocytosed they should in theory fuse with the membrane as well.

Thank you for your valuable comment. As suggested, we have investigated how well tumor cell-derived EVs interact with other types of cells in the tumor microenvironment including macrophages, fibroblasts and endothelial cells. As a result, confocal microscopic images showed that the tumor cell-derived EVs interacted more efficiently with tumor cells, compared with other types of cells, indicating their preferential localization in the tumor cells in the tumor microenvironment. We have included the data in **supplementary figure 12**.

In addition, the reason why FLs fuses more selectively with tumor cell membrane as shown in **figure 3** and **supplementary figure 11** is because we screened and optimized liposome compositions to achieve tumor cell-selective membrane fusion by varying the length and saturation level of lipid chain, and the portion of positively charge lipids (DOTAP) and PEGylated lipids (PEG-DSPE). Thus, the difference in the fusion efficacy would come from the rigidity and surface charge of liposomes. However, the precise mechanism behind the tumor cell-selective membrane fusion is still unknown and further investigation is required.

In main text,

“Furthermore, the tumor cell-derived EVs interacted more efficiently with tumor cells compared with macrophages, fibroblasts, and endothelial cells, indicating preferential localization of SR-lipids in the tumor cells in the tumor microenvironment (**Supplementary Fig. 12**).”

In supplementary figure 12,

Supplementary Figure 12. Treatment of tumor cell-derived EVs to macrophages (raw264.7 and J774A.1), fibroblasts (NIH-3T3), tumor (HeLa) and endothelial cells (HUVEC). To generate tumor cell-derived EVs containing fluorophore-lipids, HeLa cells were treated with fluorophore-FLs for 4 h and the secreted EVs were isolated using ultracentrifugation. Each type of cells was then treated with HeLa-derived EVs containing fluorophore-lipids for 2 h, and imaged with confocal microscopy. Scale bar represents 20 μm .

In supplementary methods,

“Cell treatment with tumor cell-derived EVs

To prepare tumor cell-derived EVs containing fluorophore-lipids, HeLa cells were treated with 140 μ M FLs containing fluorophore-lipids for 4 h, washed thoroughly to remove free liposomes in the media, and further incubated for 24 h. The secreted EVs were isolated from the supernatant using the ultracentrifugation protocol and resuspended in PBS. Tumor cells (HeLa), macrophages (Raw264.7 and J774A.1) or non-parenchymal cells (HUVEC and NIH-3T3) were treated with tumor cell-derived EVs containing fluorophore-lipids for 2 h and imaged with confocal microscopy (Nikon, Tokyo, Japan).”

8. Can the authors please also check for FL-EVs in serum at different time points and not only analyse the half-life on FLs.

Thank you for your valuable comment. As suggested, we have analyzed the blood samples to determine the presence of EVs packaging the fluorophore-lipids in the blood after intravenous injection of fluorophore-FLs. 5-week-old female Balb/c mice bearing 4T1 tumors were injected through the tail vein with 200 μ l of 5 μ M fluorophore-FLs. To achieve complete distribution of FLs throughout the body and subsequent production of engineered EVs, first blood extraction was performed at 24 h after FL injection. Blood samples were collected at 24 h, 48 h, and 72 h after injection. Plasma was isolated from whole blood by centrifugation at 1,000 g for 10 min at 4°C. Plasma was further centrifuged at 10,000 g for 30 min at 4°C to remove cell debris. EVs were pelleted by ultracentrifugation at 100,000 g for 2 h at 4°C and resuspended in 100 μ l PBS. The fluorescence of EV solution was measured using Gemini XPS Microplate Reader (Molecular Devices, Sunnyvale, CA, USA). As a result, the fluorescence of EVs from fluorophore-FL-treated mice was hardly measurable at all time points and showed no significant difference from that of EVs from PBS-treated control mice. The fluorescence intensity of EVs at all time points was lower than that of 0.1% of FLs in the blood immediately after injection. The value we obtained was regarded as background fluorescence as compared to control EVs.

There is a possible explanation for this result. As shown in the biodistribution data (**Supplementary Fig. 17a**), FLs accumulate mainly in liver, spleen, and tumor at 24 h post-injection. However, FLs do not effectively fuse with macrophages in liver and spleen, which hinders secretion of EVs packaging fluorophore-lipids from these cells into blood. Since FLs show tumor cell-selective membrane fusion, tumor cells would be the main cells that can effectively secrete EVs packaging fluorophore-lipids. It has been known that tumor-derived EVs circulate in blood and play various physiological functions including metastasis and immune suppression [Yang, C. & Robbins, P. D. The roles of tumor-derived exosomes in cancer pathogenesis. *Clinical & developmental immunology* **2011** (2011) 842849, and Vader, P., Breakefield, X. O. & Wood, M. J. Extracellular vesicles: emerging targets for cancer therapy. *Trends in molecular medicine* **20**, (2014) 385-393]. However, EVs secreted from the perivascular region of tumors (mainly from tumor cells) within short time periods (24 h to 72 h) after FL treatment would be significantly diluted in blood, thus lowering the possibility of their detection. Considering that EVs are successfully packaged with fluorophore-lipids only when FLs are fused effectively with tumor cell membranes and thus tumor is the only source of fluorescent EVs in blood, it was likely that we don't detect circulating tumor-derived EVs packaging fluorophore-lipids due to their extremely low concentration in blood within the time periods tested in this study.

9. Authors claim through in vitro experiments that SR lipids are not membrane localised preferentially in liver and spleen, however authors cannot correlate in vitro observations with that of in vivo. Can the authors please explain the findings in supplementary figure 14? To me it seems that these SR receptors are primarily enriched in kidney and tumour followed by liver and spleen. Is there a risk of kidney and/or liver toxicity? Is it possible for the authors to exchange the phototoxicity molecule to a toxic molecule induced by a small molecule instead and investigate the body wide effect of the treatment? Just to make sure the specificity of the treatment towards tumour tissue. This experiment is very important to perform in one way or the other to inform the readers of the applicability of the technique. Both kidney and liver toxicity is a major problem in a clinical setting and needs to be further evaluated.

Thank you for your valuable comment. Inefficient membrane localization of SR-lipids in liver and spleen can be explained with our findings in **supplementary figure 14**. Large quantities of FLs carrying SR-lipids were cleared by the mononuclear phagocytic systems of liver and spleen as shown in **supplementary figure 14a**, which is similar to conventional nanoparticles. If the biotin-lipids delivered to liver and spleen are efficiently localized on the membranes of macrophages, liver and spleen accumulations of SA injected subsequent to biotin-FLs should be enhanced. However, liver and spleen accumulations of the subsequently-injected SA were not significantly increased compared with those of SA injected alone as shown in **supplementary figure 14b**, indicating poor membrane localization of biotin-lipids in these organs.

In case of kidney, although SA injected subsequent to biotin-FLs looked like being enriched in kidney, it was not caused by the pre-accumulated biotin-lipids because the kidney accumulation of biotin-FLs was negligible as shown in **supplementary figure 14a**. Substantial accumulation of SA in kidney is mainly determined by its intrinsic pharmacokinetic property. Since SA is a protein (52.8 kDa) and its size is smaller than 10 nm, it showed high accumulation in kidney. In addition, if the biotin-lipids delivered to kidney are effectively localized on the membranes, kidney accumulation of SA injected subsequent to biotin-FLs should be enhanced. However, kidney accumulation of the subsequently-injected SA was not significantly increased compared with that of SA injected alone as shown in **supplementary figure 14b**, indicating that there was no SR(biotin)-mediated accumulation of targeted agent(SA) in kidney. **Collectively, these results demonstrate that tumor-selective membrane fusion of biotin-FLs significantly enhanced tumor-selective targeting of subsequently-injected SA (supplementary figure 14b)**. However, the application of the technique is not limited to biotin-streptavidin combination but can be extended to any receptor-ligand combination systems where selected targeted agents have more favorable pharmacokinetics for clinical translation.

Regarding the therapeutic cargo attached to SA, we chose the photosensitizer because SA bound to biotin moieties generated on the cell surface was not internalized effectively into the cell and remained on the cell surface. By taking advantage of its long retention on the cell surface, we were able to achieve enhanced membrane-specific phototherapeutic effects as observed in **figures 4a and 4c**.

In main text,

“Confocal microscopy revealed intrinsic fluorescence of Ce6, primarily on the plasma membrane at 1 h post-incubation, and the majority remained for over 5 h post-incubation

(Fig. 4a), indicating the rapid membrane localization of Ce6-SA and its minimal intracellular uptake.”

“The phototoxicity of membrane-bound Ce6-SA was significantly influenced by the dose of radiation (**Fig. 4b** and **Supplementary Fig. 13**), not by the incubation time (**Fig. 4c**), suggesting that the minimal internalization of Ce6-SA bound to the cell surface preserves the membrane-selective phototherapeutic effects.”

Toxic molecules would not be good as therapeutic cargo attached to SA in this study because they must be delivered inside the cell to induce cytotoxicity. Furthermore, using the photosensitizer, we did not observe any visible systemic toxicity and body weight change. Although we did not perform our experiments with toxic molecules due to advantages of photosensitizer mentioned above, we strongly agree that using toxic molecules instead of photosensitizer would bring a lot of benefits for the treatment of tumors, particularly the regions where light cannot penetrate. We would like to investigate it in the future work with new SR-protein combination in which the binding can induce effective internalization of proteins into the cells.

Minor concerns:

1. Author has elucidated the involvement of EVs in the transport of SR lipids very well, however most of the information are in supplementary figures, therefore author can add some of the key data in a new figure in the manuscript.

Thank you for your valuable comment. As suggested, we have included some of the key data in figures of the main text.

Reviewer #2 (Remarks to the Author):

1. A full paper rather than a letter would be preferable. A section by section demonstration of the work can guide the readers to follow the manuscript better.

Thank you for your valuable comment. As suggested, we have changes the format from a letter to a full paper.

2. Because of the need to utilize fusogenic liposome (FL) before the PDT treatment, it means it needs an additional step prior to the PDT treatment. How is the fusogenic liposome targeted to only cancer cells? The authors have not clearly demonstrated good, cancer selective, targetability of the FL. Is it possible that FL fuses into other normal epithelial cells?

Thank you for your valuable comment. As shown in **supplementary figure 17a**, FLs accumulated mainly in tumor, liver and spleen, which is similar to what conventional

nanoparticles do. Systemically-administered FLs are easily cleared by macrophages of the mononuclear phagocyte system such as liver and spleen because host's immune system recognizes them as foreign entities. Some of the FLs can accumulate in the tumor tissues by taking advantage of leaky tumor vasculature [known as an enhanced permeability and retention (EPR) effect]. Since FLs were not decorated with any tumor-targeting ligands, their *in vivo* tumor targeting (or accumulation) was mainly achieved by the EPR effect.

Regarding their selective fusion into tumor cells, we already showed that FLs fuses more selectively into tumor cells compared with other types of cells including macrophages, fibroblasts, and endothelial cells (**Fig. 3** and **Supplementary Fig. 11**). The reason of this result is because we screened and optimized liposome compositions to achieve tumor cell-selective membrane fusion by varying the length and saturation level of lipid chain, and the portion of positively charge lipids (DOTAP) and PEGylated lipids (PEG-DSPE). Thus, the difference in the fusion efficacy would come from the rigidity and surface charge of liposomes. However, the precise mechanism behind the tumor cell-selective membrane fusion is still unknown and further investigation is required.

3. Can the EVs travel outside of the tumor tissue?

Thank you for your valuable comment. As suggested, we have analyzed the blood samples to determine the presence of EVs packaging the fluorophore-lipids in the blood after intravenous injection of fluorophore-FLs. 5-week-old female Balb/c mice bearing 4T1 tumors were injected through the tail vein with 200 μ l of 5 μ M fluorophore-FLs. To achieve complete distribution of FLs throughout the body and subsequent production of engineered EVs, first blood extraction was performed at 24 h after FL injection. Blood samples were collected at 24 h, 48 h, and 72 h after injection. Plasma was isolated from whole blood by centrifugation at 1,000 g for 10 min at 4°C. Plasma was further centrifuged at 10,000 g for 30 min at 4°C to remove cell debris. EVs were pelleted by ultracentrifugation at 100,000 g for 2 h at 4°C and resuspended in 100 μ l PBS. The fluorescence of EV solution was measured using Gemini XPS Microplate Reader (Molecular Devices, Sunnyvale, CA, USA). As a result, the fluorescence of EVs from fluorophore-FL-treated mice was hardly measurable at all time points and showed no significant difference from that of EVs from PBS-treated control mice. The fluorescence intensity of EVs at all time points was lower than that of 0.1% of FLs in the blood immediately after injection. The value we obtained was regarded as background fluorescence as compared to control EVs.

It has been known that tumor-derived EVs circulate in blood and play various physiological functions including metastasis and immune suppression [Yang, C. & Robbins, P. D. The roles of tumor-derived exosomes in cancer pathogenesis. *Clinical & developmental immunology* **2011** (2011) 842849, and Vader, P., Breakefield, X. O. & Wood, M. J. Extracellular vesicles: emerging targets for cancer therapy. *Trends in molecular medicine* **20** (2014) 385-393]. However, we did not detect circulating tumor-derived EVs packaging fluorophore-lipids because EVs secreted from the perivascular region of tumors (mainly from tumor cells) within short time periods (24 h to 72 h) after FL treatment would be significantly diluted in blood.

4. The FL will most likely be toxic at higher concentration. Only when tumors are large (>1 cc), does the tumor microenvironment form and thus deep tumor targeting becomes important, while the authors studied an only 50 mm³ sized tumor. As there may be only limited amounts of FL that can be introduced into the animal, would that work for the large tumors for which this new treatment is envisioned. Although theoretically it would be possible for the FL to modify the entire tumor, the volume of the tumor increases at a much faster ratio than the surface area of the tumor. Do the authors believe that enough EVs can be created to cover a bigger tumor volume?

Thank you for your valuable comment. To address this concern, we have performed additional experiments to investigate if FLs can deliver SR-lipids throughout a larger tumor with the same injection dose as one used in this study. We prepared small (62.5 mm³; L=5 mm, W=5 mm) and large (360 mm³; L=10 mm, W=8.5 mm) 4T1 tumors and compared fluorophore-lipid distribution in both tumors. Mice bearing 4T1 tumors were intravenously co-injected with 200 μ l of 5 μ M fusogenic (FLs) and PEGylated liposomes (PLs) incorporating fluorophore-lipids. At 24 h post-injection, tumors were collected for histological analysis. As a result, the fluorophore-lipids delivered by FLs were more uniformly distributed throughout both small and large tumors compared with those delivered by PLs. Thus, we believe that membrane-specific delivery of SR-lipids by FLs can lead to their uniform distribution even in larger tumors via EV-mediated intercellular transfer. We have included the data in **supplementary figure 15**.

In main text,

“Uniform distribution of fluorophore-lipids delivered by FLs was also observed in relatively large tumors (~ 360 mm³) (**Supplementary Fig. 15**).”

In supplementary figure 15,

Supplementary Figure 15. Distribution of fluorophore-lipids delivered by FLs or PLs in 4T1 tumors with difference size. (a and b) Representative confocal microscopic images of tumor sections from the 4T1 tumors with estimated volumes of 62.5 mm³ (a) and 360 mm³ (b) collected at 24 h after intravenous co-injection of FLs (magenta) and PLs (green) incorporating fluorophore-lipids. Nuclei were stained with Hoechst (blue). Scale bars represent 1 mm.

In supplementary methods,

“In vivo fluorescence tissue imaging

To observe SR-lipid delivery in small and large tumors, tumor models were generated by implanting 5×10^5 4T1 cells in 5-week-old female Balb/c mice. When the tumor volume reached approximately 60 or 360 mm³, mice were co-injected with 200 μ l of 5 μ M FLs and PLs containing fluorescent lipids. After 24 h, the mice were sacrificed for histological analysis. The tumor sections were then examined under a confocal microscope (Nikon).”

5. The comparison with conventional targeted therapy (supplementary figure 17) is an important piece of data which should be included into the main figure.

Thank you for your valuable comment. As suggested, we have included the data showing the comparison with conventional targeted agents in figure 5c.

Reviewers' comments:

Reviewer #1 (Remarks to the Author):

I would like to thank the authors for the extensive work performed in this revised manuscript. The authors has addressed nearly all concerns raised and I am overall very happy with the manuscript. However, before acceptance there are a few small revisions needed in order to further improve the quality of the paper and meet the standards of Nature communications.

-Supplementary fig 8: EM pics need to be shown in a zoomed out format as well. See International Society of Extracellular vesicles (ISEV) guidelines for further information.

-Supplementary fig 8b. Comassie blue stain reveals a lot of protein contamination in the EV preparations. This is most likely albumin. This should be commented.

- Supplementary figure 7. ImmunoEM needs a zoomed out picture. Also cannot show a cloud of dark spots only, one needs to see both un-labelled and labelled particles in the same picture.

Reviewer #2 (Remarks to the Author):

The revisions are satisfactory and the paper is now good for publication.

Reply to reviewer' comments

We have addressed the reviewer' comments as follows:

Reviewer #1 (Remarks to the Author):

1. Supplementary fig 8: EM pics need to be shown in a zoomed out format as well. See International Society of Extracellular vesicles (ISEV) guidelines for further information.

Thank you for your valuable comment. As suggested, we have retaken TEM images and included zoomed-out images together with enlarged images in supplementary figure 8a.

In supplementary figure 8a,

Supplementary Figure 8. Morphology, protein profile and size of EVs after biotin-FL treatment.
(a) Representative TEM images of EVs from cells treated with either PBS (control) or biotin-FLs. Scale bars indicate 100 nm.

2. Supplementary fig 8b. Coomassie blue stain reveals a lot of protein contamination in the EV preparations. This is most likely albumin. This should be commented.

Thank you for your valuable comment. To reduce protein contamination, we have redone the coomassie blue stain after washing EVs using a 100K Amicon centrifugal filter. As a result, the band indicating albumin was significantly reduced. We have included this new data in supplementary figure 8b.

In supplementary figure 8b,

Supplementary Figure 8. Morphology, protein profile and size of EVs after biotin-FL treatment.
(b) Coomassie blue staining of protein extracts from EVs from cells treated with either PBS (control) or biotin-FLs.

3. Supplementary figure 7. ImmunoEM needs a zoomed out picture. Also cannot show a cloud of dark spots only, one needs to see both un-labelled and labelled particles in the same picture.

Thank you for your valuable comment. As suggested, we have retaken immuno-TEM images with streptavidin-nanogold (10 nm). In new images, we clearly observed gold nanoparticles bound on the membrane of biotin-EVs while we hardly detected gold nanoparticles on the membrane of control EVs. We have included zoomed-out immuno-TEM images together with enlarged images in supplementary figure 7.

In supplementary figure 7,

Supplementary Figure 7. Representative TEM images of unmodified EVs and biotin-EVs after streptavidin-nanogold staining. EVs from the cells treated with either PBS (control) or biotin-FLs were stained with streptavidin-nanogold(10 nm). EVs from biotin-FL-treated cells showed binding of gold nanoparticles on the surface while EVs from PBS-treated cells did not. Scale bars indicate 100 nm.

REVIEWERS' COMMENTS:

Reviewer #1 (Remarks to the Author):

The authors has addressed all the concerns raised and the manuscript is ready to be published.